# The Penis, the Vagina and HIV Risk: Key Differences (Aside from the Obvious)

**DOI:** 10.3390/v14061164

**Published:** 2022-05-27

**Authors:** Rupert Kaul, Cindy M. Liu, Daniel E. Park, Ronald M. Galiwango, Aaron A. R. Tobian, Jessica L. Prodger

**Affiliations:** 1Departments of Medicine and Immunology, University of Toronto, Toronto, ON M5S 1A8, Canada; rupert.kaul@utoronto.ca; 2Department of Medicine, University Health Network, Toronto, ON M5S 1A8, Canada; 3Department of Environmental and Occupational Health, Milken Institute School of Public Health, George Washington University, Washington, DC 20052, USA; cindyliu@email.gwu.edu (C.M.L.); danpark@email.gwu.edu (D.E.P.); 4Rakai Health Sciences Program, Kalisizo P.O. Box 279, Uganda; rmgaliwango@rhsp.org; 5Department of Pathology, Johns Hopkins University School of Medicine, Johns Hopkins University, Baltimore, MD 21205, USA; atobian1@jhmi.edu; 6Department of Microbiology and Immunology, Schulich School of Medicine and Dentistry, Western University, London, ON N6A 5C1, Canada; 7Department of Epidemiology and Biostatistics, Schulich School of Medicine and Dentistry, Western University, London, ON N6A 5C1, Canada

**Keywords:** HIV, foreskin, penis, vagina, genital immunology, microbiota

## Abstract

Globally, most Human Immunodeficiency Virus type 1 (HIV) transmission occurs through vaginal–penile sex (heterosexual transmission). The local immune environment at the site of HIV exposure is an important determinant of whether exposure during sex will lead to productive infection, and the vaginal and penile immune milieus are each critically shaped by the local microbiome. However, there are key differences in the microbial drivers of inflammation and immune quiescence at these tissue sites. In both, a high abundance of anaerobic taxa (e.g., *Prevotella*) is associated with an increased local density of HIV target cells and an increased risk of acquiring HIV through sex. However, the taxa that have been associated to date with increased risk in the vagina and penis are not identical. Just as importantly, the microbiota associated with comparatively less inflammation and HIV risk—i.e., the optimal microbiota—are very different at the two sites. In the vagina, *Lactobacillus* spp. are immunoregulatory and may protect against HIV acquisition, whereas on the penis, “skin type” flora such as *Corynebacterium* are associated with reduced inflammation. Compared to its vaginal counterpart, much less is known about the dynamics of the penile microbiome, the ability of clinical interventions to alter the penile microbiome, or the impact of natural/induced microbiome alterations on penile immunology and HIV risk.

## 1. Introduction

The incidence of Human Immunodeficiency Virus type-1 (HIV) remains far above the 2020 targets established by Joint United Nations Programme on HIV/AIDS (UNAIDS) [1], and there are concerns that ripple effects of the coronavirus disease of 2019 (COVID-19) pandemic on HIV treatment and prevention programs may lead to an increase in HIV incidence [2]. Globally, most of the 1.5 million new infections with HIV in 2020 were attributable to unprotected sex (vaginal or anal) with an HIV-infected partner who was not being treated with effective antiretroviral therapy (ART). Although the prevalence of HIV is increased almost 30-fold in men who have sex with men [3], most global HIV transmission occurs within heterosexual populations, particularly in sub-Saharan Africa (SSA) which is home to two-thirds of people living with HIV (PLWH), and where females accounted for 63% of new HIV infections in 2020. 

The field has acquired a quite sophisticated understanding of the role that vaginal and penile immunology play in HIV susceptibility, with inflammation at each tissue site disrupting epithelial integrity and recruiting highly susceptible CD4+ T cells [4]. More recently, multiple lines of research have demonstrated that although the vaginal and penile immune milieu are each critically shaped by the local microbiome, the microbial drivers of inflammation and immune quiescence at these tissue sites appear to be quite distinct. Our goal in this review is to explore similarities and differences between the immune and microbial determinants of HIV susceptibility in the penis and female genital tract.

## 2. Sexual HIV Transmission

In simple terms, an individual’s risk of acquiring HIV through sex is dependent on two major factors: (i) viral exposure, comprised of both the frequency of unprotected sex with an HIV-infected partner and the level of virus in that person’s genital/rectal secretions; and (ii) the type mucosal surface(s) that is/are exposed to virus during sex and its biological parameters, which further alter the probability of productive HIV infection after exposure [5]. With regards to viral exposure, despite many advances in our understanding of genital HIV transmission, several fundamental issues remain unresolved, such as whether transmission is primarily mediated via cell-free or cell-associated virus (or both) (reviewed in [6]). Both are present in the genital secretions of an infected person, and both are reduced on effective antiretroviral treatment; while cell-free HIV RNA levels correlate closely with in vivo transmission risk, cell-associated virus is transmitted more effectively in some explant models, meaning that this issue remains unresolved. However, well-powered studies have clearly demonstrated that there is no risk of sexual HIV transmission when an individual’s plasma viral load is undetectable (i.e., <200 RNA copies/mL) [7,8,9]. The knowledge that undetectable = untransmissible (U = U) has transformed global treatment guidelines and the lives of individuals living with HIV and their partners. 

The focus of this review is how the type and biological parameters of the exposed mucosa influence the likelihood that HIV is transmitted. Due to gaps in the care cascade (an individual knowing their HIV status, being able to access stable treatment, and attaining an undetectable viral load), sexual transmission of HIV continues. This review focuses on understanding biological risk factors relevant to the sexual acquisition of HIV and is informed by human cohort studies that have HIV infection as an endpoint, rather than in vitro or explant studies. Our hope is that this will inform best practices and novel prevention modalities. 

The increased HIV incidence among women within heterosexual epidemics suggests that the cervicovaginal mucosa is more susceptible to HIV than the penile mucosa. However, while a meta-analysis of multiple studies demonstrated that the per-exposure probability of male-to-female HIV transmission was approximately double that of female-to-male [10], additional factors need to be considered. Anal sex is common within heterosexual communities but is under-reported due to stigma [11]; since the anorectal mucosa is much more susceptible to HIV than that of the vagina or penis [12], this could increase per-act HIV risk for the female partner. Patterns of sexual partnering in many areas of SSA mean that HIV-uninfected young women may be more likely to have an older male sexual partner [13], increasing the risk that he is HIV infected. Other important factors that alter female HIV risk include the increased susceptibility of the adolescent genital mucosa, which has a higher proportion of columnar epithelium and elevated proinflammatory cytokines compared to older women [14]; the fact that women are disproportionately affected by sexual violence, which is associated with increased HIV transmission risk [15]; and the younger acquisition in women of HIV-associated sexually transmitted infections such as HSV-2 [16]. Furthermore, penile circumcision reduces HIV risk [17], and in communities that do not traditionally perform penile circumcision the per-exposure risk of HIV acquisition tends to be higher in men than women [18]. Therefore, it is unclear whether the increased per-exposure risk observed in women is truly driven by an increased biological susceptibility of the cervico-vaginal mucosa, or by these and other important factors. 

Most of our knowledge of the biological factors that alter an individual’s susceptibility to HIV infection after exposure during penile–vaginal sex has been derived from studies of the female genital tract. While it may be tempting to extrapolate findings from the female genital tract to the penis, there are substantial differences between these tissue sites, including epithelial structure, immunology, microbiome, and other biological risk factors. Furthermore, even in the absence of classical STIs, there are notable similarities and differences in the vaginal and penile bacteria that have now been associated with modifying host susceptibility to HIV infection. This review will focus on the immunology and microbiome of the male and female genital tract (Box 1), particularly as they relate to HIV susceptibility in heterosexual HIV transmission. 

Box 1Penis vs. Vagina: similarities and differences in HIV susceptibility.
*Similarities*
Per-coitus risk of HIV transmission is comparable between sites.Both are stratified squamous epithelia.Immune activation is associated with increased risk, and quiescence with protection.Antimicrobial peptides with anti-HIV activity are associated with increased risk.STIs increase inflammation and HIV risk.Genital washing increases inflammation and HIV risk.Abundances of *Prevotella* spp. correlate with inflammation and HIV risk.

*Differences*
Epithelial structure: the penis has a cornified outer layer and no goblet cells.On the penis, different STIs infect and inflame different sites: HSV-2, HPV, and chancroid affect the skin, while chlamydia, gonorrhea, syphilis and trichomoniasis affect the urethra.In the vagina, microbiota dominated by *Lactobacillus* spp. are associated with lower risk of HIV; on the penis, microbiotas with low anaerobe abundance are associated with lower risk.


## 3. Genital Immunology and HIV Risk

### 3.1. Epithelial Structure

The vaginal mucosa and penile skin (foreskin, glans, and shaft) are continually renewing multi-layer stratified epithelia that express high levels of various keratins. However, these epithelia differ in several key features. The outermost layers of both epithelia are comprised of terminally differentiated keratinocytes that are enucleated, but on the penis, these cells are additionally cornified. The cornified cells of penile skin are tightly connected by corynedesmosomes and limit water loss through the extrusion of lamellar bodies to form an intracellular lipid envelope. In contrast, the outermost cells of the vaginal epithelium are loosely connected and lack adaptations to limit water loss [19,20,21]. The vagina is bathed in mucus from goblet cells present in the columnar epithelium of the cervix in the upper reproductive tract (endocervix). The outermost vaginal epithelial cells also contain high levels of glycogen, an important carbon source for *Lactobacillus* spp., while desquamation of the cornified penile skin frees lipids and fatty acids [19,22,23,24]. The vaginal epithelium of reproductive-age females is also thicker than that of penile skin (100–20 μm vs. 70–100 μm) [25,26].

### 3.2. HIV Target Cells

Despite differences in the microstructure of the epithelium, local inflammation clearly increases the risk of acquiring HIV in both the penis and vagina [27,28,29]. In vitro and non-human primate studies have revealed that inflammation can increase HIV risk through multiple pathways. First, inflammation can increase target cell density. Sexual transmission of HIV occurs exclusively through immune cells expressing CCR5 and CD4, and in non-human primates, the mucosal density of CCR5+ cells is a key determinant of whether exposure to SIV will result in infection [30,31]. In addition to increasing target cell density, inflammation may also recruit immune cell populations that are more permissive to infection than others. Generally, activated immune cells are more permissive to infection and produce more virus once infected [32,33,34]. Some CD4 T cell subsets are particularly susceptible to HIV, including CCR6+ Th17 cells, which are present in both the penile and vaginal epithelia, and CCR10+ Th22 cells, which are highly abundant in skin [35,36,37,38,39,40,41]. T cell subsets bearing specific patterns of integrins, such as α_4_β_7_ and α_4_β_1_, are more permissive to HIV infection, and their density in the mucosa correlates with risk in non-human primates [34,42,43]. In addition to T cells, local inflammation can also recruit macrophages and recruit and activate dendritic cells in the mucosa, which can be directly infected by HIV or facilitate CD4 T cell infection [44,45].

### 3.3. Genital Cytokines

Human studies directly linking HIV acquisition to genital mucosal immune cell density or phenotype have not been performed due to cost and logistical challenges. However, the concentrations of soluble cytokines on the epithelial surface correlate with vaginal and penile immune cell densities [29,46,47,48], and these surface cytokines may be collected and stored relatively easily for future nested studies (e.g., penile skin swabs, cervicovaginal lavage, etc.). Two large trials of HIV prevention modalities in sub-Saharan Africa, one at the Rakai Health Sciences Program (RHSP trial of penile circumcision [49]) and one at the Centre for the AIDS Program of Research in South Africa (CAPRISA 004 trial of vaginal tenofovir gel [50]), collected such genital secretions longitudinally and nested studies have been performed to examine the association between pre-existing genital cytokines and the risk of acquiring HIV. 

The consistent association between genital chemotactic cytokines concentrations and HIV suggests that inflammation facilitates HIV infection in both the penis and vagina. Among females [27], the chemotactic cytokines MIP-1α, MIP-1β, IL-8, and IP-10 (out of IL-1α, IL-1β, IL-6, IL-7, IL-8, IL-10, GM-CSF, IP-10, MCP-1, MIP-1α, MIP-1β, TNF) were independently associated with HIV risk. Other cytokines, especially IL-1α and GM-CSF, were also shown to contribute to risk after accounting for the total number of elevated vaginal cytokines. In males [29], chemotactic cytokines IL-8 and MIG (out of IL-1α, IL-8, MCP-1, MIG, MIP-3α, RANTES, and GM-CSF) were significantly associated with HIV acquisition (note MIG was not measured in the vagina and MIP-1α, MIP-1β, and IP-10 were not measured in the penis). Despite the low level of cytokines (IL-1α, MCP-1, MIP-3α, RANTES, and GM-CSF were detected in <10% of participants) on the penile surface, the total number of elevated cytokines also correlated with HIV risk. None of the cytokines measured to date, on either the penile or vaginal surfaces, have correlated negatively with the risk of HIV acquisition (i.e., increased levels of the cytokine associated with decreased risk of HIV). 

### 3.4. Innate Antimicrobial Peptides

In contrast, the role of innate antimicrobial peptides (AMP) in HIV susceptibility is less clear. Two classes of AMP have been shown to have in vitro anti-HIV activity: cationic peptides [human neutrophil peptides 1–4 (HNP 1–4); human β-defensins 1–3 (HBD 1–3); LL-37] and anti-proteases [secretory leukocyte protease inhibitor (SLPI); elafin and its precursor, trappin-2] [51,52,53,54,55,56,57,58,59,60,61,62,63,64,65,66,67,68,69]. However, high in vivo vaginal concentrations of HNP1-3 and LL-37 were associated with a significantly *increased* risk of acquiring HIV [70]. Sub-preputial concentrations of HNP1-3 and SLPI were associated with an increased risk of subsequent HIV acquisition after controlling for other risk factors [71]. In contrast to the vagina, LL-37 levels on the penis were not associated with increased risk of HIV acquisition, but they also were not associated with protection. The difference between in vitro and in vivo findings suggests that AMP may have a dual nature: in addition to their direct antimicrobial activity, AMPs also act as immune signaling molecules and can induce pro-inflammatory changes in the epithelium [51,72] and may thus indirectly increase HIV susceptibility. The increased in vivo HIV risk associated with AMPs suggests that the detrimental inflammation likely outweighs the beneficial antiviral activity. 

### 3.5. Individuals Who Are HIV-Exposed but Remain SeroNegative (HESN)

As prospective studies linking tissue parameters to HIV seroconversion are logistically difficult, an alternative approach is to identify mucosal correlates of protection. Studies of individuals who have been highly exposed to HIV but remained seronegative (HESN) aim to identify unique attributes of mucosal protection. Such observational studies of high-risk individuals were performed prior to the demonstrated efficacy of PrEP (pre-exposure prophylaxis) and TasP (treatment as prevention) and are often no longer ethical. Vaginal immune cells isolated from HESN sex workers have a lower frequency of HIV target cells (CCR5+CD4+), produce less pro-inflammatory cytokines, and express lower levels of pattern recognition receptors (PRR) as compared to new sex workers [73,74,75]. Similar protection from HIV is afforded by an immune quiescent mucosa on the penis. In a study of HIV discordant couples, men who remained seronegative despite many years of unprotected sex with a viremic female partner had a reduced proportional abundance of foreskin Th17 cells and CD4+ cells producing the pro-inflammatory cytokine TNF compared to unexposed men [76].

In summary, there are many commonalities between the penis and vagina in the role of genital immune activation in HIV susceptibility. Despite distinct differences in the epithelial barrier structure of penile skin and the vaginal mucosa, immune activation is associated with an increased risk of infection at both sites, and immune quiescence is associated with protection. The importance of immune activation in in vivo HIV susceptibility is underscored by prospective studies demonstrating that high levels of AMP are associated with increased HIV risk in both the penis and vagina, despite the ability of AMP to neutralize HIV in vitro. The development of HIV prevention modalities should closely consider local inflammatory effects on vaginal and penile tissue before moving to in vivo trials.

## 4. Clinical Factors That Alter Genital Immunology and Enhance Risk

### 4.1. Genital Infections

While it is clear that mucosal inflammation enhances HIV susceptibility in both the male and female genital tracts [4], the causes of these immune changes are much better defined in the vagina. In the vagina, bacterial STIs such as *Neisseria gonorrhoeae* and *Chlamydia trachomatis* both increase levels of cervicovaginal cytokines and recruit increased numbers of activated CD4+ T cells to the endocervix [77], and *Trichomoniasis vaginalis* infection (trichomoniasis) has similar effects [78,79]. Common sexually transmitted viral infections can also reactivate in the genital tract and cause substantial immune changes. This is particularly true of herpes simplex virus type 2 (HSV-2) infection, which not only induces a long-lasting infiltration of virus-specific CD4+ T cells at the site of herpetic ulcers [80] but is also associated with generalized increases in activated endocervical CD4+ T cells even in the absence of virus reactivation [46,81], likely because the host immune response against HSV2 is predominantly CD4-mediated [82]. However, despite these STI-HIV associations, it is important to note that neither bacterial STI prevention nor HSV-2 suppression has been able to reduce HIV acquisition or transmission in clinical trials [83].

Cytomegalovirus (CMV) is also frequently shed in female genital secretions [84], and while the impact of this virus on HIV risk is not well understood, genital reactivation has been linked to increased levels of IL6 and TNF [85]. Interestingly, although human papillomavirus (HPV) infection is extremely common in sexually active women and has been epidemiologically linked to increased risk of HIV acquisition, neither prevalent HPV infection nor host immune clearance induces dramatic immune changes [86]. However, HPV clearance was associated with increased Langerhans cells and prevalent HPV infection with modest elevations in some vaginal chemokines [86,87]. Other microbial causes of vaginal discharge that are not sexually transmitted include bacterial vaginosis (BV) and vulvovaginal candidiasis (VVC); BV has important effects on HIV risk and genital immunology that are discussed later, and VVC is also associated with increased HIV acquisition risk [88,89] although the impact of VVC on genital immunology is less clear [78].

The impact of STIs on penile immunology is broadly similar to that in the female genital tract, but it is important to consider the differential effects of STIs on the immunology of the foreskin and urethra; the former is thought to be the site of most penile HIV acquisition in uncircumcised men, and the latter in circumcised men. Urethral STIs such as chlamydia increase inflammatory cytokines in semen [90] with minimal effects on foreskin immunology [91], and semen reactivation of CMV in semen increases the number of semen activated CD4+ T cells [92] without altering cytokine levels [93]. However, STIs that involve skin, particularly HSV2 infection, not only increase the density of CD4+ T cells in the foreskin [94] but also their activation level and expression of the HIV coreceptor CCR5 [95]; these effects are in addition to the ulcerations caused by herpes and syphilis that directly enhance HIV access to target cells in the submucosa. In addition, HPV clearance has been associated with HIV seroconversion in men and increased penile epidermal dendritic cells [96].

### 4.2. Genital Hygiene and Products

Later sections in this review will discuss the important impact of the genital microbiome on the immunology of the penis and vagina, and it is likely that clinical parameters can enhance HIV risk both via direct mucosal effects and/or via an impact on the genital microbiome. This is especially true for genital washing: vaginal washing with soap or other alkaline substances is associated with an increased incidence of both bacterial vaginosis (BV) and HIV [97], and penile washing immediately after sex tended to be associated with increased HIV acquisition [98] although effects on the microbiome have not been defined. The application of more toxic compounds to the mucosa likely mediates HIV risk via direct epithelial disruption, and this is likely the reason for the increased HIV risk associated with the application of the spermicide nonxynol-9 (N9) [99], which induces epithelial damage to both the vaginal and rectal mucosa [100,101], although the penile impact of this spermicide has not been defined. The vaginal application of sexual lubricants with high osmolality can also damage the epithelium [102,103], although a link between HIV acquisition and specific lubricants has not been described to our knowledge, and again, there have been no penile studies. 

### 4.3. Contraception

The only user-directed contraceptive used by men is the penile condom, whose barrier function directly reduces HIV risk and whose impacts on penile immunology or [104,105] the penile microbiome have not been studied to our knowledge. In contrast, several contraceptive choices employed by women have important immune effects on the female genital tract. The intra-uterine device (IUD) has its contraceptive effect through the induction of uterine inflammation [106], and so it is perhaps not surprising that the IUD also induces cervicovaginal inflammation as well as increases the incidence of BV [107]. The effects of hormonal contraceptives on the microbiome, genital immunology, and HIV risk are the subject of ongoing debate, and a detailed discussion is beyond the scope of this review. However, oral (estrogen-based) contraceptives have generally not been linked to immune changes or HIV risk [108], while progestin-based contraceptives have been linked to HIV acquisition in some studies [109] but not others [110], and to induce an increase in activated cervicovaginal CD4+ T cells in some studies [111] but not others [105]. 

### 4.4. Penile Circumcision

The final important clinical difference between penile and vaginal HIV acquisition risk is that the former can be substantially reduced through a widely available and safe surgical procedure, namely penile circumcision. This procedure reduces HIV risk not only through the direct removal of HIV-susceptible tissues but also by altering the immune and microbial environment of the remaining tissues [17]. Specifically, penile circumcision leads to a decrease in HSV-2 and HPV in men and HPV, BV and *Trichomonas vaginalis* in women and also a chronic decrease in the concentration of inflammatory cytokines on the coronal sulcus [29,112,113,114,115], and also induces an aerobic tissue environment that leads to a rapid and near-complete loss of pro-inflammatory anaerobic penile bacteria from the coronal sulcus. Interestingly, these effects are specific to the coronal sulcus since penile circumcision induces minimal change in the immunology of the urethra despite a moderate reduction in anaerobic bacteria at this site [116]. 

## 5. Penile and Vaginal Microbiome in HIV Risk

The genital microbiota has recently been identified as an important risk factor in HIV heterosexual transmission. Having a high abundance of specific genital anaerobic bacteria has been associated with >4-fold increased risk of HIV acquisition in both the vagina [117,118] and penis [119,120] in longitudinal studies that directly assess HIV seroconversion. While there are major distinctions in penile and vaginal microbiome composition, there are some notable similarities in how penile and vaginal bacteria have been linked to susceptibility to HIV infection.

The vaginal microbiota is optimally dominated by one or more species of *Lactobacillus* (*L. crispatus*, *L. jensenii*, *L. gasseri*, *L. iners*), which inhibit the growth of pathogenic organisms by producing lactic acids, bacteriocins, and biosurfactants and by maintaining a low pH in the vaginal lumen [121,122,123,124,125,126,127,128,129,130]. When *Lactobacillus* spp. are not dominant, the vagina is colonized by a diverse set of strict and facultative anaerobes (e.g., *Gardnerella*, *Megasphera*, *Prevotella*, *Snethia*, *Atopobium*; Figure 1), referred to as “molecular” bacterial vaginosis (BV) [131]. Molecular BV is associated with a multitude of adverse sexual and reproductive health outcomes, including symptomatic BV (e.g., abnormal discharge, itching, malodor, elevated pH), increased susceptibility to bacterial STIs (e.g., *N. gonorrhoeae*, *C. trachomatis*), and gynecologic and obstetrics outcomes [47,132,133,134,135,136,137,138,139,140,141,142,143,144,145]. A nested case-control study of young South African women (FRESH cohort, cases *n* = 31 and controls *n* = 205) found molecular BV to be associated with a >4-fold increased risk of HIV acquisition compared to females with vaginal microbiota dominated by *L. crispatus* (HR = 4.4, 95% CI = 1.17–16.61, *p* = 0.028), an association independent of other HIV risk factors, including *C. trachomatis* [118]. There is evidence that this increased HIV risk is driven, at least in part, by immune activation and the concomitant recruitment of CCR5+CD4+ and Th17 cells. In animal and in vitro studies, vaginal microbiota observed during molecular BV has been shown to impair vaginal epithelial maturation, increase mucosal HIV target cell density, and induce changes in epithelial barrier function [118,146]. In vivo, molecular BV is associated with increased pro-inflammatory cytokines in the vaginal lumen and a 17-fold increase in local HIV target cells [118,146]. However, we are only beginning to understand the interaction between vaginal bacteria, the vaginal epithelium, and HIV, and it is likely that other mechanisms may contribute to bacteria-driven HIV risk (e.g., direct interactions between bacteria and HIV, disruption of epithelial barrier function, etc.). 

In contrast to the vaginal microbiota, *Lactobacillus* spp. are relatively rarely observed on the penis (coronal sulcus). Instead, penile microbiotas range from those with a high bacterial load, generally with a predominance of strict anaerobes, to those with a low overall bacterial load, which tend to contain fewer strict anaerobes and a higher proportional abundance of skin-associated facultative anaerobes such as *Corynebacterium* and *Staphylococcus* [120,148]. The high bacterial load/anaerobe abundance microbiota is common on the uncircumcised penis and is comprised of a diverse mix of Gram-negative and positive strict anaerobes, which overlap to some degree with molecular BV (e.g., *Prevotella*, *Dialister*, *Anaerococcus*). However, there are also clear distinctions that include a high abundance of *Peptoniphilus*, *Porphyromonas*, *Peptostreptococcus*, and *Peptoniphilaceae* and a low prevalence/abundance of *Gardnerella*, *Snethia*, *Megasphera*, and *Apotobium* (Figure 1). The low bacterial load/low anaerobe profile is rare when the coronal sulcus is covered by a foreskin but is the dominant microbiota of circumcised men [148], which is notable given that penile circumcision confers significant protection against HIV [49,149,150]. Indeed, a nested case-control study of uncircumcised males found that uncircumcised males with a *Prevotella*-dominated penile microbiota had a >5-fold increased odds of acquiring HIV during the 2-year study period compared to men with low *Prevotella* abundance (quartile 4 vs. 1: aOR = 5.42, 95%CI = 1.63–21.19) when adjusting for age, number of sex partners and condom use [119]. Much like the vagina, a high abundance of strict anaerobes is associated with increased penile cytokines and local HIV target cells [119,120]. Therefore, it is likely that the protective mechanism of penile circumcision against HIV acquisition is mediated at least in part by the elimination of many strict anaerobes from the penile microbiota [148].

While a high abundance of strict anaerobes is associated with inflammation and HIV risk for both the penis and vagina, not all taxa associated with risk in the vagina are relevant to the penis and vice versa (Figure 2). Three studies have begun to tease out the taxa that may be driving HIV risk from amongst the diverse and frequently co-occurrent taxa on the uncircumcised penis and during molecular BV in the vagina [117,118,119]. Six penile bacterial species from three genera have been associated with increased risk of HIV seroconversion, as well as increased levels of local IL-8 and α-defensins and increased density of Th17 and CCR5+/CD4+ T cells: *Peptostreptococcus anaerobius*, *Prevotella bivia*, *Prevotella disiens*, *Dialister propionicifaciens*, *Dialister micraerophilus*, and a genetic near neighbor of *Dialister succinatiphilus*. These six species have been named Bacteria Associated with Seroconversion and Immune Cells, or BASIC [119]. In the vagina, HIV acquisition has been associated with an increased abundance of *Prevotella*, *Veillonella*, *Mycoplasma*, *Sneathia*, *Parvimonas*, *Gemella*, *Eggerthella*, and *Megasphaera* [117,118]. These lists must be interpreted cautiously because anaerobes in the genital microbiota are highly co-occurring, and it is difficult to discern from observational studies which taxa are truly responsible for local inflammation and increased HIV risk. However, these data do suggest a higher abundance of strict anaerobes is associated with HIV acquisition in both the penis and vagina, albeit potentially with some differences in groups of anaerobic bacteria. Further studies in more diverse geographic regions and populations (i.e., outside sub-Saharan Africa), combined with both in-depth microbial analysis and mechanistic in vitro studies, are needed to fully elucidate the genital bacteria that increase HIV risk at the two sites.

Despite their important role in vaginal health, there is no evidence to date that lactobacilli play a role in penile health or protection against HIV, while in the vagina, an increased proportional (but not absolute) abundance of *Lactobacillus* spp. has been associated with HIV protection [117,118]. In the vagina, the ability of lactobacilli to protect against HIV may be direct, for instance, through pH-mediated inactivation of HIV or metabolite-mediated immunomodulatory effects that protect against HIV infection, or indirect, through the exclusion of inflammatory BV-associated bacteria that increase HIV risk. The latter is supported by the observation that BV treatment in women induced a very rapid drop in genital inflammatory cytokines and chemokines that was specifically linked to a decrease in the absolute abundance of BV-associated bacteria rather than to any absolute increase in *Lactobacillus* spp. [151].

## 6. Implications and Future Studies

Although we are beginning to understand the immune correlates of HIV susceptibility in the foreskin, much less is known about other penile sites—most notably the urethra, which is assumed to be the site of penile HIV acquisition in most circumcised (and a subset of uncircumcised) men. No study has yet linked HIV acquisition with urethral immune parameters, and exploring this question will require large longitudinal cohort studies with HIV as an outcome, perhaps nested in the context of a clinical trial.

There are still many knowledge gaps regarding the genital microbiota and the mechanism(s) by which they influence mucosal immunology and HIV susceptibility. Natural variation and dynamics of the genital microbiota and local immunology have been explored to some degree in the female genital tract but very little in the penis, so there is a need to understand variation across the lifespan (e.g., adolescence) and between geographical regions/populations. The determinants of heterosexual transmission of genital microbiota components, and species/strain-level differences in bacteria shared between the penis and vagina, might also be explored. HIV-focused future research should determine which specific genital bacteria cause increased risk and identify relevant virulence mechanisms. The link between the microbiota and HIV risk at other mucosal tissue sites exposed to HIV during sex, such as the urethra or rectum, also remains to be investigated. Ideally, such studies inform clinical interventions to eliminate specific pathogenic bacteria while leaving ‘optimal’ microbiota components intact. As discussed above, different approaches may be necessary for the penis and vagina since the components of a low-risk microbiota are quite different at these two sites. Better in vitro models that can recapitulate the genital epithelium and support relevant bacterial communities will be pivotal in answering these important questions.

In summary, there is considerable overlap in the immune and microbial correlates of HIV risk on the penis and in the vagina, but also key differences that must be considered when designing new modalities to prevent heterosexual transmission of HIV. While inflammation undoubtedly increases tissue HIV susceptibility at both sites, the surfaces of the penile and vaginal epithelia differ in key structural properties and in the microbial drivers of inflammation and immune quiescence, and this must be considered when trying to shape a mucosal environment that is resilient to HIV.

## Figures and Tables

**Figure 1 viruses-14-01164-f001:**
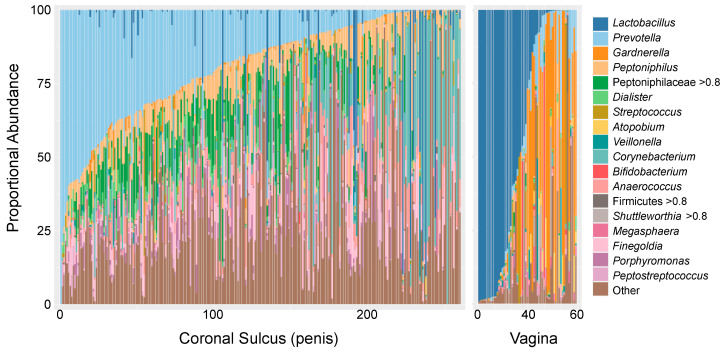
Proportional abundances of the 18 most prevalent taxa on the coronal sulcus of the uncircumcised penis (*n* = 260 [119]) and in the vagina (*n* = 60 [147]) of Rakai Cohort participants. Participants of each sex are ordered based on the abundance of the most prevalent taxa at that site: *Prevotella* on the coronal sulcus and *Lactobacillus* in the vagina.

**Figure 2 viruses-14-01164-f002:**
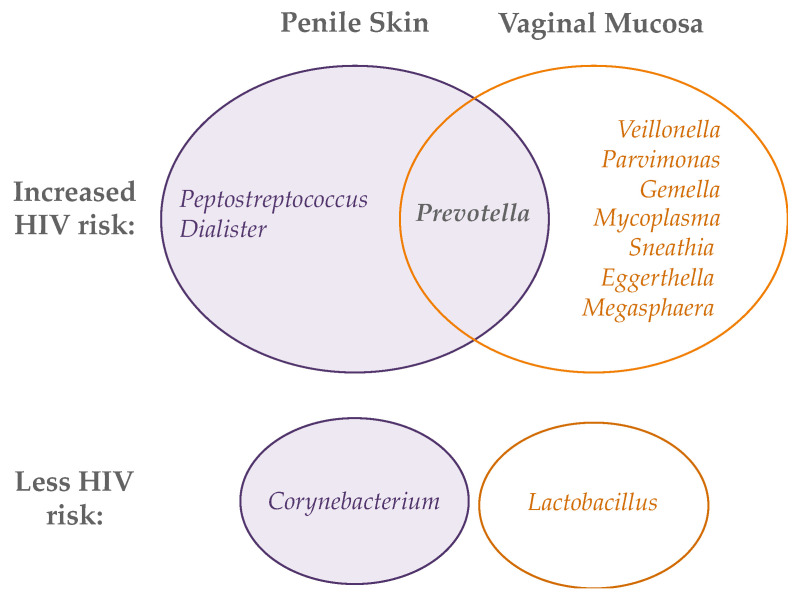
Bacterial genera whose abundance on the penis (**left**) and vagina (**right**) is associated with either increased (**top** Venn diagram) or decreased (**bottom**) HIV acquisition.

## Data Availability

Not applicable.

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
