# Peer review of "The Penis, the Vagina and HIV Risk: Key Differences (Aside from the Obvious)"

_viruses, 2022, doi:10.3390/v14061164_

Round 1

Reviewer 1 Report

The present manuscript is a review aiming to highlight the differences between penile and vaginal microbiota in the context of HIV transmission.

The paper is in fact an impressive expert review, exhaustive but easy to read, well-sourced, and which takes in consideration all the biases reported so far regarding this topic.

The presence of Corynebacterium spp. in the "less HIV group" (Figure 2 and in the text box 1) can suggest that it is a beneficial microbe in this context (that is a priori not the case), but just the reflect of a "low anaerobe" penile microbiota profile. This is however clearly explained in the body text of the manuscript. I believe that the authors should perhaps in the text box emphasize that the penile microbiota dominated by anaerobes is associated with higher risk rather than that dominated by Corynebacterium is at lower risk.

Line 276: Tichomonas vaginalis should be italicized

My compliments to the authors for this work.

Author Response

We thank the reviewer for their appreciation of our manuscript.

We have now changed the Text Box to read "In the vagina, microbiota dominated by Lactobacillus spp. are associated with lower risk of HIV; on the penis, microbiotas with low anaerobe abundance are associated with lower risk."

We have corrected the italicization of Trichomonas vaginalis.

Reviewer 2 Report

In this manuscript, Kaul and colleagues review the existing knowledge of HIV risk factors in males and females. Some of the topics discussed include the differences in epithelial cells of the penis vs. the vagina, effect of microbiome on susceptibility, and immune activation/cytokine composition. The manuscript is thorough and well written, and only minor suggestions are listed below.

  1. Line 69 states that the adolescent genital mucosa is more susceptible to HIV infection. While a thorough explanation of this concept is beyond the scope of this manuscript, it would be interesting to elaborate on the cause, if known.
  2. A venn diagram of the cytokine composition associated with male/female susceptibility to HIV (similar to Fig. 2) would be helpful
  3. Section 3.2 could be expanded/or clarified by including a list of cells and receptors used by HIV for entry. This could be in the form of a table, for example.

Author Response

We thank the reviewer for their time and helpful comments.

  1. We have added that the adolescent mucosa “has a higher proportion of columnar epithelium and elevated proinflammatory cytokines compared to older women” (lines 90 – 92) but agree that wider discussion may be beyond the scope of the review.
  2. We agree with the reviewer that such a Venn diagram would be informative, however, unfortunately, there is insufficient overlapping data of the cytokines measured in the penis and vagina to construct one. This is partially driven by the investigators of the two studies choosing to examine different cytokines, and partially due to the relatively low quantity of secretions that can be collected from the surface of the penis – only those cytokines present at high concentration could be associated with HIV susceptibility, yet this does not necessarily indicate that those cytokines measured but detected in too few participants to be statistically associated with HIV susceptibility are not relevant. We have done our best to describe this in the body of the text.
  3. While we have added some detail to the text, including the susceptibility of genital macrophages, (line 195) listing other susceptible cell types from other tissues not involved in sexual HIV transmission (e.g., microglia, etc.) may distract from the focus of the review.

Reviewer 3 Report

A well-constructed review, which makes a very clear point of the situation. 
I have three suggestions to make this review more comprehensive:
- transmission via the urethral epithelium is quickly mentioned throughout the manuscript. It seems to me that it should be mentioned at the beginning of the manuscript because it represents a major issue for circumcised people and for the understanding of the problem. Moreover, it has an implication in the establishment of the HIV reservoir (see the work of M. Bomsel's group). 
- The inoculum is also quickly described. The risk of HIV sexual transmission in relation to the viral load in secretion seems to be an important point in the context of the U=U paradigm.
- Finally, add few words on the type of transmission, either by cell-free virus or by cell-associated virus. This point has to my knowledge never been properly addressed, but it has its place in this review as a gap in our knowledge.

Minor points:
- problem with the symbol font, line 104; 120; 353
- Line 147 "No cytokines from ..." of course out of those tested!

Author Response

We thank the reviewer for their appreciation of the manuscript and their excellent suggestions.

  1. We agree with the reviewer may be an important site for HIV acquisition in both circumcised and uncircumcised men. As stated in the “Implications and future studies” section, this will be an important area for future research. However, this review has been informed by human cohort studies with HIV acquisition as an endpoint. This has been clarified in the introduction on lines 71 – 78.
  2. Thank you for the suggestion of highlighting the importance of viral load. We have now added the following paragraph to section 2. Sexual Transmission of HIV:

Well powered studies have clearly demonstrated that there is no risk of sexual HIV transmission when an individual’s plasma viral load is undetectable (i.e., <200 copies/ml; refs: Rodger et al., JAMA, 2018; Rodger et al., Lancet 2019; Bavinton et al., Lancet, 2018). The knowledge that undetectable = untransmissible (U=U) has transformed global treatment guidelines and the lives of individuals living with HIV and their partners. However, due to gaps in the care cascade (an individual knowing their HIV status, being able to access stable treatment, and attaining an undetectable viral load), sexual transmission of HIV continues. This review focuses on understanding biological risk factors relevant to sexual acquisition of HIV, with the hope of informing best practices and novel prevention modalities.

  1. We have now added the following lines on this interesting topic:

However, despite many advances in our understanding of genital HIV transmission, several fundamental issues remain unresolved, such as whether transmission is primarily mediated via cell-free or cell-associated virus (or both) [reviewed in Cavarelli M et al, 2020]. Both are present in the genital secretions of an infected person, and both are reduced on effective antiretroviral treatment; while cell-free HIV RNA levels correlate closely with in vivo transmission risk, cell-associated virus is transmitted more effectively in some explant models, meaning that this issue remains unresolved. “

  1. Thank you for noting these typos.
    1. The symbols have been corrected
    2. The following clarification has been made: None of the cytokines measured to date, on either the penile or vaginal surfaces, have correlated negatively with risk of HIV acquisition (i.e., increased levels of the cytokine associated with decreased risk of HIV).